# Learning to Describe for Predicting Zero-shot Drug-Drug Interactions

**Fangqi Zhu**[1,3][*]**, Yongqi Zhang** [4][†]**, Lei Chen** [5]**, Bing Qin**[1]**, Ruifeng Xu**[1,2,3][†]

[1] Harbin Institute of Technology, Shenzhen, China
[2] Peng Cheng Laboratory, Shenzhen, China
[3] Guangdong Provincial Key Laboratory of Novel Security Intelligence Technologies
[4] 4Paradigm Inc., Beijing, China
[5] Hong Kong University of Science and Technology (GZ), Guang Zhou, China
zhufangqi.hitsz@gmail.com, yzhangee@connect.ust.hk, xuruifeng@hit.edu.cn

## Abstract

Adverse drug-drug interactions (DDIs) can compromise the effectiveness of concurrent drug administration, posing a significant challenge in healthcare. As the development of new drugs continues, the potential for unknown adverse effects resulting from DDIs becomes a growing concern. Traditional computational methods for DDI prediction may fail to capture interactions for new drugs due to the lack of knowledge. In this paper, we introduce a new problem setup as zero-shot DDI prediction that deals with the case of new drugs. Leveraging textual information from online databases like DrugBank and PubChem, we propose an innovative approach TextDDI with a language model-based DDI predictor and a reinforcement learning (RL)-based information selector, enabling the selection of concise and pertinent text for accurate DDI prediction on new drugs. Empirical results show the benefits of the proposed approach on several settings including zero-shot and few-shot DDI prediction, and the selected texts are semantically relevant. Our code and data are available at `https://github.com/zhufq00/DDIs-Prediction`.

## 1 Introduction

Adverse drug-drug interactions (DDIs) pose a significant challenge in modern healthcare as they can alter the effectiveness of drugs when administered concurrently (Edwards and Aronson, 2000; Yan et al., 2021). With the continuous development of drugs each year, the potential for unknown adverse effects resulting from DDIs has become a growing concern, often leading to increased risks or even severe heart failure. Given the constrained understanding of these newly developed drugs and the high cost associated with conducting comprehensive clinical trials, there is a pressing need to explore cost-effective and efficient approaches, such

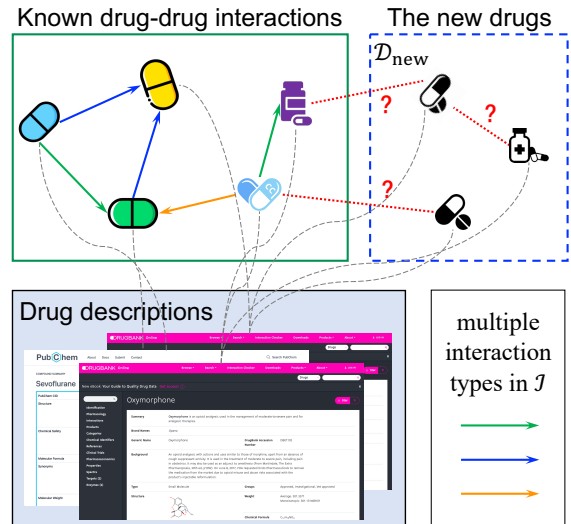

Figure 1: A graphical illustration of zero-shot DDI prediction based on the textual descriptions. The green box in the upper left contains known interactions between a set of known drugs. The blue box in the upper right includes several new drugs that are not seen during training. The task is to use drug descriptions from websites to predict interactions for the new drugs.

as computational methods, to identify potential DDIs (Percha and Altman, 2013; Han et al., 2022; Margaret Savitha and Pushpa Rani, 2022).

The existing computational methods, which learn experience from a large number of known interactions, can be classified into two categories, i.e., pure-DDI-based and knowledge graph (KG)-based, according to the resources they use. The pure-DDI-based methods directly use Fingerprints of molecules (Rogers and Hahn, 2010) or learn embeddings (Yao et al., 2022) to predict the interaction between drugs. These methods are shallow and work under the assumption that the new interactions have similar patterns as the existing ones. KG-based methods leverage external KG (e.g. HetioNet (Himmelstein and Baranzini, 2015)), a large biomedical network representing the relationships between drugs and proteins, diseases, pathways,

---

[*]Work done when F. Zhu was interned at 4Paradigm.
[†]Corresponding authors.

etc., to support the DDI prediction (Karim et al., 2019; Vashishth et al., 2020; Zitnik et al., 2018; Lin et al., 2020; Yu et al., 2021). KG embedding techniques (Zhang et al., 2022b) and graph neural networks (GNNs) (Kipf and Welling, 2016; Zhang and Yao, 2022) are commonly used in these methods. Considering that not all drugs can have a connection with the external KG, these methods may fail for drugs with rare biomedical connections.

New drug development is increasing (Atanasov et al., 2021), but predicting DDIs before clinical tests remain challenging for most computational methods due to a limited understanding of new drugs. In this paper, we study the scenario of DDI prediction on new drugs, i.e., zero-shot DDI prediction, with limited interaction knowledge. Benefiting from the online databases like DrugBank[1] and PubChem[2] that provide knowledge on drugs, we can have access to many textual resources describing every single drug, as graphically illustrated in Figure 1. The materials encompass several aspects of drugs, such as backgrounds, indications, and mechanism of action. Therefore, we aim to learn from these texts and capture essential information for DDI prediction on the new drugs.

However, there come two main challenges in using textural information for DDI prediction. First, the descriptions are professional with many special tokens and expressions. Therefore, the general large language models (LLMs) like ChatGPT and GPT4[3] are inadequate in dealing with this scenario. Additionally, the texts are lengthy and contain noise, leading to elevated computational expenses and heightened training complexities for LMs. To address these challenges, we design a specific prompt for the DDI prediction task and fine-tune a pre-trained language model to capture the domain knowledge, and then utilize reinforcement learning (RL) to describe the drugs with shorter but more relevant text for each drug pair.

Our approach, named TextDDI, consists of two modules: an LM-based DDI predictor and an RL-based information selector. With the specially designed task prompt, the LM-based DDI predictor is first trained to predict interactions based on randomly sampled drug descriptions, effectively capturing domain-specific knowledge. To appropriately describe the drugs, we design an RL-based

information selector to generate drug descriptions for each drug pair as its prompt. Specially, the generated drug descriptions are conditioned on drug pairs to improve flexibility. By learning from the reward returned from the DDI predictor, the information selector effectively selects suitable sentences that align with the given pairs of drugs. The main contributions can be summarized as follows:

- **Problem:** We propose a novel problem setup that leverages textual descriptions for DDI prediction, especially on new drugs as a zero-shot setting.
- **Method:** We carefully design an LM-based DDI predictor with an RL-based information selector that accurately predicts DDIs with short and relevant descriptions.
- **Experiments:** We achieve better performance in zero-shot, and few-shot DDI predictions with the proposed method. In addition, we show that the selected texts by the information selector are semantically relevant to the target prediction.

## 2 Preliminaries

### 2.1 Problem Formulation

DDIs prediction aims to predict the interaction type given a pair of drugs simultaneously used. During training (in the green box), there are several interactions between a set of known drugs. During inference (in the blue dashed box), we predict the interaction between a new drug with a known drug, or between two new drugs. Figure 1 provides a graphical illustration of the proposed problem. Formally, we define this problem as follows:

**Definition 1 (Zero-shot DDI prediction)** *Let* $\mathcal{D}$ *be the set of all drugs,* $\mathcal{D}_{new}$ *be the set of new drugs, and* $\mathcal{I}$ *be the set of interaction types. Formally, the zero-shot DDIs prediction task is to learn a mapping* $\mathcal{M} : (\mathcal{D}_{new} \times \mathcal{D}) \cup (\mathcal{D} \times \mathcal{D}_{new}) \to \mathcal{I}$ *where* $\mathcal{M}$ *maps a drug pair* $(u, v) \in ((\mathcal{D}_{new} \times \mathcal{D}) \cup (\mathcal{D} \times \mathcal{D}_{new}))$ *to the corresponding interaction type* $i \in \mathcal{I}$.

### 2.2 Background

**A shift from symbolic methods** The previous symbolic methods (Rogers and Hahn, 2010; Yao et al., 2022; Zitnik et al., 2018; Yu et al., 2021; Lin et al., 2020; Zhang et al., 2023) rely on drug-drug interactions or the drugs' connection with other biomedical entities to encode each drug as an embedding. However, it lacks efficiency in encoding drugs due to the lack of knowledge of the newly developed drugs. To tackle this issue, we shift from symbolic interactions to textual information

---

[1] https://go.drugbank.com
[2] https://pubchem.ncbi.nlm.nih.gov
[3] https://chat.openai.com/ and https://openai.com/gpt-4, respectively.

describing drugs. Textual information is crucial in enabling the model to comprehend the background of new drugs, thereby facilitating accurate predictions of potential interactions between new drugs and other drugs.

For every drug $u \in \mathcal{D}$ with name $u^{\text{name}}$, we can obtain its raw descriptions $u^{\text{raw}}$ from either the Drugbank or PubChem database. Our selection of information includes the drug's background, indications, mechanism of action, etc. However, these descriptions tend to be extensive and cluttered with extraneous data. When combining the descriptive text from two drugs, the resulting text often has thousands of tokens, with only a small portion being actually pertinent to our task. Therefore, the primary goal of our study is to extract concise and relevant text from the provided drug descriptions, and utilize it to correctly predict the DDI.

**Large language models**   The large language models (LLMs), such as ChatGPT and GPT4, have demonstrated remarkable proficiency across many tasks, including machine translation (Jiao et al., 2023), text summarization (Bang et al., 2023), and question answering (Tan et al., 2023), etc. Despite their successes, ChatGPT still lags behind baseline encoder models such as RoBERTa (Liu et al., 2019) in certain specialized tasks, including affective text classification(Amin et al., 2023) and event extraction(Gao et al., 2023; Zhu et al., 2023a,b). We conducted an evaluation of ChatGPT and GPT4 in our zero-shot DDI prediction task, following the instruction style employed by Gao et al. (2023). The results (provided in Appendix A) indicate that the general LLMs struggle to accomplish zero-shot DDI prediction task due to their lack of relevant knowledge, suggesting the necessity of our research.

**Prompt learning**   Recent studies suggest that LMs can efficiently conduct new tasks through a few in-context demonstrations (Brown et al., 2020), but their performance is significantly influenced by the formulation of the prompt (Zhao et al., 2021; Lu et al., 2022). To address this sensitivity, several methods for automatic prompt generation have been proposed. In the continuous space, Lester et al. (2021) introduces prompt-tuning, which adds trainable tokens as the prompt for each task to improve performance. In the discrete space, RLPrompt (Deng et al., 2022) and TEMPERA (Zhang et al., 2022a) propose to use RL to adjust the prompt.

However, RLPrompt only generates task-specific prompts, while TEMPERA is designed to adjust the order or content of given inputs. Both of them cannot address the problem of selecting sentences over a long and sample-dependent raw input.

**Reinforcement learning for text selection**   The prompt generating of drug pairs can be considered as a kind of text selection problem. Within this category, Yin et al. (2022) employed reinforcement learning for weakly-supervised paraphrase generation. They used reinforcement learning to select valuable samples from a large set of weakly labeled sentence pairs for fine-tuning the paraphrase generation pre-trained language models. Lee and Lee (2017) utilized reinforcement learning to select sentences for text summarization tasks, using ROUGE-2 score as the reinforcement learning's supervisory signal. However, both of them are unable to address the task of generating sentence sequences related to drug-drug interaction (DDI) prediction for drug pairs in the absence of existing supervision signals.

## 3   Method

To predict zero-shot DDI using textual information, our methodology is designed to generate shorter but more relevant prompts from lengthy and noisy drug descriptions and predict DDIs based on these generated prompts. The drug description we used includes information about the drug, such as its indication, mechanism of action, and more. The original description can be several thousand tokens long, the distribution of lengths is provided in Appendix B, but we limit the length of the generated prompt for each drug pair to only a few hundred tokens, e.g., 256 tokens. Our method, named TextDDI, consists of two modules: an LM-based DDI predictor and an RL-based information selector. The DDI predictor predicts the type of drug-drug interaction based on the specially designed prompt of the drug pair, while the RL-based information selector is responsible for generating the corresponding prompt based on the drug descriptions of the drug pair. Additionally, we alternate the training pipeline of these two modules to facilitate their mutual enhancement. Figure 2 illustrates an overview of our method in the inference stage.

### 3.1   DDI Predictor

The DDI predictor is designed to accurately predict DDIs by utilizing prompts generated from drug de-

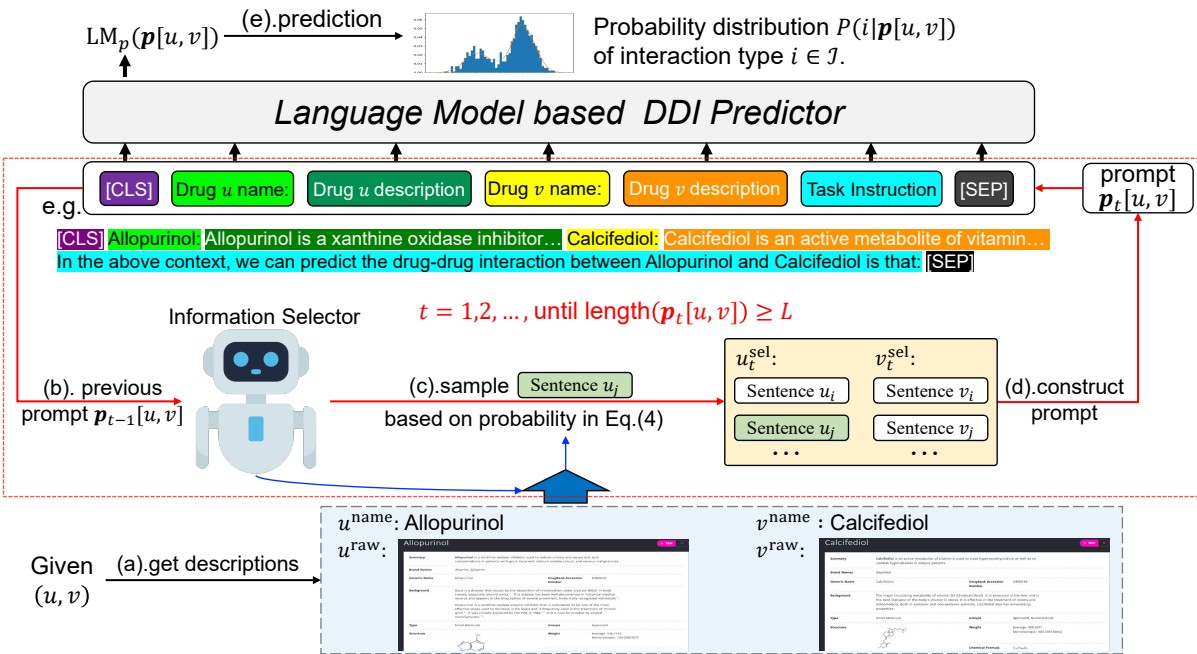

Figure 2: The inference procedure of the proposed TextDDI. (a). Given a drug pair, we obtain the drug descriptions from websites. (b). The previous prompt $\boldsymbol{p}_{t-1}[u,v]$ (where $\boldsymbol{p}_0[u,v]$ is "[CLS] $u^{name}$ : {} $v^{name}$ : {} [Task Instruction] [SEP]") is fed into the selector. (c). The information selector samples one sentence from descriptions based on their priority score. (d). The prompt $\boldsymbol{p}_t[u,v]$ is constructed with $\boldsymbol{p}_{t-1}[u,v]$ and the new sentence $u_j$. (e). When the prompt $\boldsymbol{p}_t[u,v]$ exceeds the length limit $L$, we stop the loop and use $\boldsymbol{p}_{t-1}[u,v]$ as the final prompt $\boldsymbol{p}[u,v]$ to predict the interaction type of drug pair $(u,v)$.

scriptions. As the prompts are important to downstream tasks (Zhao et al., 2021; Lu et al., 2022), we design a prompt for the DDI prediction task here. As graphically illustrated in the top part of Figure 2, the prompt contains names, and descriptions of given drug pairs $(u,v)$. To guide the model to do DDI prediction, the task instruction is given as in the blue content. Hence, the prompt is designed to indicate the task, and describes the information of both drugs to be predicted.

However, considering that the drugs' description can have thousands of tokens, while the complexity of the pre-trained LM increases quadratically in terms of sequence length, we cannot use all the descriptions. Instead, we randomly select some sentences and put them into the prompts. For a given drug pair $(u,v)$, we can have their names $u^{name}$ and $v^{name}$, and we randomly select sentences $u^{sel}$ and $v^{sel}$ from the descriptions $u^{raw}$ and $v^{raw}$, respectively. Afterwards, we create the prompt $\boldsymbol{p}[u,v]$, constrained with a maximum length limit of $L$, by combining [CLS], $u^{name}$, $u^{sel}$, $v^{name}$, $v^{sel}$, the task instruction, and [SEP].

After constructing the prompt $\boldsymbol{p}[u,v]$, we proceed to train a classification model, referred to as the DDI predictor, to predict the interaction

type $i \in \mathcal{I}$ between $u$ and $v$. The classification model is based on a pre-trained language model $\mathrm{LM}_p(\cdot)$, whose input is $\boldsymbol{p}[u,v]$ and the output embedding of the final hidden state of the [CLS] token is passed through a multi-class linear classifier to predict the interaction type. For a certain example $(u,i,v) \in \mathcal{S}_{\mathrm{tra}}$, the classification model returns the probability $P(i|\boldsymbol{p}[u,v])$, and the DDI predictor is trained by minimizing the cross-entropy loss:

$$\mathcal{L}_p = \sum_{(u,i,v) \in \mathcal{S}_{\mathrm{tra}}} -\log\left(P(i|\boldsymbol{p}[u,v])\right). \quad (1)$$

### 3.2 Information Selector

To create succinct and relevant descriptions that serve as prompts for drug pairs, a straightforward approach would be to annotate these descriptions and then train a model to select them, but this requires costly biomedical experts. In addition, there is no simple heuristic to indicate such relevance. Therefore, we propose a learning method that discovers which kind of information is helpful for DDI prediction. Since the sentences are discrete, we design an innovative reinforcement learning (RL) framework in this part to automatically generate drug-pair-dependent prompts, with the DDI predictor serving as the supervisory signal.

**Reinforcement learning formulation.** Considering the sentences describing drugs have some correlation with each other, we formulate the prompt generation process as a Markov Decision Process (MDP) (Bellman, 1957). A 4-tuple $(\mathcal{S}, \mathcal{A}, \mathcal{P}, \mathcal{R})$ is defined to represent the MDP, where $\mathcal{S}$ is the state space, $\mathcal{A}$ contains all available actions, $\mathcal{P}$ is the policy designed to select actions, and $\mathcal{R}$ is the reward function based on the performance of DDI prediction.

**State space $\mathcal{S}$.** Forming an MDP, the prompt is generated sequentially to select the relevant information related to DDI prediction. Denote $\boldsymbol{p}_t[u, v]$ as the prompt generated at step $t$, we directly use $\boldsymbol{p}_t[u, v]$ as the state. Recall in Section 3.1, the prompt is composed of drug-related information, the task descriptions, and some auxiliary tokens. Specifically, we have

$$\mathcal{S} : \boldsymbol{p}_t[u, v] = [\text{CLS}] \ u^{\text{name}} : u_t^{\text{sel}} \ v^{\text{name}} : v_t^{\text{sel}}$$
$$[\text{Task instruction}] \ [\text{SEP}]. \quad (2)$$

At the initial step $t = 0$, we have $u_0^{\text{sel}} = v_0^{\text{sel}} = \emptyset$. And at each time step $t$, we select one sentence from either $u^{\text{raw}}$ or $v^{\text{raw}}$ and adds it into $u_{t-1}^{\text{sel}}$ or $v_{t-1}^{\text{sel}}$, respectively. If the length of $\boldsymbol{p}_t[u, v]$ surpasses the predefined maximum limit $L$, the process will be halted and $\boldsymbol{p}_{t-1}[u, v]$ will be returned as the final prompt $\boldsymbol{p}[u, v]$.

**Action space $\mathcal{A}_t$.** To generate a prompt based on the description of a drug pair, we include the sentence sequences $u^{\text{raw}}$ and $v^{\text{raw}}$ from the raw description for drug pair $(u, v)$ in our action space. In order to avoid the selection of repeated sentences, we take out one sentence from the non-selected ones. Consequently, our action space

$$\mathcal{A}_t = u^{\text{raw}} \cup v^{\text{raw}} - u_t^{\text{sel}} \cup v_t^{\text{sel}} \quad (3)$$

gradually narrows down as step $t$ increases.

**Policy $\mathcal{P}(a_t \in \mathcal{A}_t | \boldsymbol{p}_{t-1}[u, v])$.** Given the prompt $\boldsymbol{p}_{t-1}[u, v]$, how to select the next sentence $a_t$ from action space $\mathcal{A}_t$ is the key component in our design. We utilize the weights of $\text{LM}_p$ in DDI predictor to initialize an identical encoder $\text{LM}_e$. The new encoder $\text{LM}_e$ is used to encode the sentences in $\mathcal{A}_t$ and measure their plausibility given prompt $\boldsymbol{p}_{t-1}[u, v]$. At step $t$, we encode the prompt $\boldsymbol{p}_{t-1}[u, v]$ by taking the output embedding of the [CLS] token $\text{LM}_e(\boldsymbol{p}_{t-1}[u, v])$ as its representation. We also encode $\text{LM}_e(a)$ for each sentence $a \in \mathcal{A}_t$.

Subsequently, a Multilayer Perceptron (MLP) is employed to evaluate the score for each possible action, denoted by

$$s(a) = \text{MLP}\big(\text{LM}_e(a) \| \text{LM}_e(\boldsymbol{p}_{t-1}[u, v])\big),$$

where $\|$ is the concatenation operation. We then apply the softmax function to derive the probability distribution over these actions $a \in \mathcal{A}_t$, i.e.,

$$\mathcal{P}(a | \boldsymbol{p}_{t-1}[u, v]) = \frac{\exp(s(a))}{\sum_{a' \in \mathcal{A}_t} \exp(s(a'))} \quad (4)$$

and sample a sentence $a_t \sim \mathcal{P}(a | \boldsymbol{p}_{t-1}[u, v])$ at step $t$. $a_t$ is inserted into $\boldsymbol{p}_{t-1}[u, v]$ to form $\boldsymbol{p}_t[u, v]$, denoted as $\boldsymbol{p}_t[u, v] := \boldsymbol{p}_{t-1}[u, v] \oplus a_t$.

**Reward $\mathcal{R}_t$.** The key to reward design is ensuring to accurately reflect the gain from $\boldsymbol{p}_{t-1}[u, v]$ to $\boldsymbol{p}_t[u, v]$. Following RLPrompt (Deng et al., 2022), we measure the quality of current prompt $\boldsymbol{p}_t[u, v]$ by the difference between the probability of the correct interaction type and the probability of the most probable incorrect type, denoted as: $d(i, \boldsymbol{p}_t[u, v]) := \log P(i | \boldsymbol{p}_t[u, v]) - \max_{i' \neq i} \log P(i' | \boldsymbol{p}_t[u, v])$. Based on $d(i, \boldsymbol{p}_t[u, v])$, the quality score is defined as

$$q(i, \boldsymbol{p}_t[u, v]) = \lambda_1^{\mathbb{C}} \lambda_2^{1-\mathbb{C}} d(i, \boldsymbol{p}_t),$$

where $\mathbb{C} := \mathbb{1}[d(i, \boldsymbol{p}) > 0]$ indicating whether the interaction type $i$ is correctly predicted, and $\lambda_1, \lambda_2 > 0$ are weights for the correct and incorrect predictions, respectively. In general, the quality score is positive when the prediction is correct, otherwise, it is negative. Then, the difference between the quality scores of $\boldsymbol{p}_t[u, v]$ and $\boldsymbol{p}_{t-1}[u, v]$ is used to measure the reward of action $a_t$,

$$\mathcal{R}_t = q(i, \boldsymbol{p}_t[u, v]) - q(i, \boldsymbol{p}_{t-1}[u, v]). \quad (5)$$

### 3.3 Training Pipeline

Our training pipeline is detailed in Algorithm 1. To capture the domain knowledge, we generate random prompts for drug pairs in the training set and minimize Eq.(1) to fine-tune the DDI predictor in line 2. The weight of $\text{LM}_p$ in DDI predictor is also used to initialize the weights of $\text{LM}_e$ in information selector in line 2. Subsequently, we repeatedly use the information selector to generate prompt and alternatively update the policy network as well as the DDI predictor in lines 3-15. For each sample $(u, i, v)$ in the train set $\mathcal{S}_{\text{tra}}$, we iteratively select

**Algorithm 1** Training Pipeline of TextDDI.

**Require:** DDI predictor, information selector, training set $\mathcal{S}_{\text{tra}}$, validation set $\mathcal{S}_{\text{val}}$.
1: fine-tune DDI predictor with randomly generated prompts in $\mathcal{S}_{\text{tra}}$ by minimizing (1);
2: initialize the weights of $\text{LM}_e$ by $\text{LM}_p$;
3: **repeat**
4:    **for** $(u, i, v) \in \mathcal{S}_{\text{tra}}$ (in mini-batch) **do**
5:        $t \leftarrow 1$ and set prompt $\boldsymbol{p}_0[u, v]$;
6:        **repeat**
7:            sample $a_t \sim \mathcal{P}(a \in \mathcal{A}_t | \boldsymbol{p}_{t-1}[u, v])$;
8:            $\boldsymbol{p}_t[u, v] \leftarrow \boldsymbol{p}_{t-1}[u, v] \oplus a_t$;
9:            compute reward $\mathcal{R}_t$ with Eq.(5);
10:           $t \leftarrow t + 1$;
11:       **until** length($\boldsymbol{p}_t[u, v]$) $> L$;
12:       update parameters of information selector by maximizing Eq.(6);
13:   **end for**
14:   update DDI predictor using the prompts generated by information selector in $\mathcal{S}_{\text{tra}}$;
15: **until** performance on $\mathcal{S}_{\text{val}}$ is stable;
16: **return** DDI predictor, information selector.

and formulate a drug-pair prompt until the length exceeds $L$ in lines 6-11. The internal rewards collected in line 9 are used to update the information selector by maximizing the expectation

$$\mathcal{J} = \sum_{(u,i,v) \in \mathcal{S}_{\text{tra}}} \sum_{t=1} \mathbb{E}_{a \sim \mathcal{P}(a | \boldsymbol{p}_{t-1}[u,v])} [\gamma^t \mathcal{R}_t], \quad (6)$$

where $\gamma < 1$ is the discount factor, in line 12 in a mini-batch manner. In line 14, we update the DDI predictor with the prompts generated by information selector such that it can be improved by the prompts with more relevant information. When the evaluated performance on validation set $\mathcal{S}_{\text{val}}$ no longer gets improved, we stop the loop and return the DDI predictor and the information selector.

# 4 Experiments

## 4.1 Experimental Setup

### 4.1.1 Datasets split for zero-shot DDI

Following (Zitnik et al., 2018; Yu et al., 2021), we conduct experiments on two benchmark datasets: DrugBank (Wishart et al., 2018) and TWO-SIDES (Tatonetti et al., 2012). To construct the dataset for zero-shot DDIs prediction, we divided the set of drugs, $\mathcal{D}$, into three disjoint sets, $\mathcal{D}_{\text{tra}}$, $\mathcal{D}_{\text{val}}$ and $\mathcal{D}_{\text{tst}}$, in **chronological order** based on the date the drugs are developed. Denote the total

number of interactions as $\mathcal{S} = \{(u, i, v) : u, v \in \mathcal{D}, i \in \mathcal{I}\}$. Based on the drug split, the train, valid, test sets are given as
- $\mathcal{S}_{\text{tra}} = \{(u, i, v) \in \mathcal{S} : u, v \in \mathcal{D}_{\text{tra}}\}$;
- $\mathcal{S}_{\text{val}} = \{(u, i, v) \in \mathcal{S} : (u \in \mathcal{D}_{\text{tra}} \cup \mathcal{D}_{\text{val}}) \wedge (v \in \mathcal{D}_{\text{tra}} \cup \mathcal{D}_{\text{val}}) \wedge (u, i, v) \notin \mathcal{S}_{\text{tra}}\}$;
- $\mathcal{S}_{\text{tst}} = \{(u, i, v) \in \mathcal{S} : (u \in \mathcal{D}_{\text{tra}} \cup \mathcal{D}_{\text{tst}}) \wedge (v \in \mathcal{D}_{\text{tra}} \cup \mathcal{D}_{\text{tst}}) \wedge (u, i, v) \notin \mathcal{S}_{\text{tra}}\}$.

In general, we ensure that the new drugs remain invisible during training. The statistics of the two zero-shot datasets are provided in Table 1.

For the DrugBank dataset, we utilized text descriptions from the Backgrounds, Indications, and Mechanism of Action sections. For the TWO-SIDES dataset, we used text descriptions from the Description section. For both datasets, we did not use the Drug Interaction section to prevent information leakage.

Table 1: The statistics for both datasets used for the zero-shot setting.

| Datasets | $|\mathcal{D}|$ | $|\mathcal{I}|$ | $|\mathcal{S}_{\text{tra}}|$ | $|\mathcal{S}_{\text{val}}|$ | $|\mathcal{S}_{\text{tst}}|$ |
|---|---|---|---|---|---|
| DrugBank | 1,710 | 86 | 129,522 | 29,402 | 26,240 |
| TWOSIDES | 604 | 200 | 30,832 | 3,962 | 5,712 |

### 4.1.2 Baselines

We compare the proposed method with three kinds of baseline methods. First, the pure-DDI based methods that direct predict $i$ given $(u, v)$, including
- MLP (Rogers and Hahn, 2010) that predicts DDIs based on drugs' fingerprints (molecular-related features);
- MSTE (Yao et al., 2022) that is a KG embedding technique to predict DDI by learning drug's and interaction's embeddings.

Second, the KG-based methods that leverage external KG to predict the interaction, including
- KG-DDI (Karim et al., 2019) that uses KG embedding technique to learn embeddings over both the DDI interactions and the external KG;
- CompGCN (Vashishth et al., 2020), Decagon (Zitnik et al., 2018) and KGNN (Lin et al., 2020) that design variants of GCN to aggregate drugs' representations from external KG for DDI prediction;
- SumGNN (Yu et al., 2021) that extracts subgraph covering given drug pair from the KG and predicts interaction based on the subgraph.

Third, the text-based methods that use textural information, including a DDI predictor with only

Table 2: Comparison of different methods on the zero-shot DDI prediction task. The reported results include the average and standard deviation across five distinct random seeds. For all the metrics, the larger values indicate better performance. The best values are in boldface and the second best are underlined.

| Datasets | | **DrugBank** | | | **TWOSIDES** | | |
|---|---|---|---|---|---|---|---|
| Type | Methods | F1-Score | Accuracy | Kappa | PR-AUC | ROC-AUC | Accuracy |
| Pure-DDI | MLP (Rogers and Hahn, 2010) | $28.7_{\pm1.1}$ | $44.9_{\pm0.9}$ | $31.7_{\pm1.0}$ | $71.0_{\pm1.2}$ | $73.1_{\pm1.4}$ | $67.1_{\pm1.5}$ |
| | MSTE (Yao et al., 2022) | $9.4_{\pm0.3}$ | $45.5_{\pm0.2}$ | $34.2_{\pm0.1}$ | $64.6_{\pm0.1}$ | $69.9_{\pm0.1}$ | $63.4_{\pm0.1}$ |
| KG-based | KG-DDI (Karim et al., 2019) | $23.5_{\pm0.4}$ | $48.8_{\pm0.9}$ | $37.9_{\pm0.6}$ | $73.6_{\pm0.3}$ | $75.8_{\pm0.4}$ | $67.8_{\pm0.2}$ |
| | CompGCN (Vashishth et al., 2020) | $30.9_{\pm1.3}$ | $49.8_{\pm0.6}$ | $40.1_{\pm0.8}$ | $72.9_{\pm0.4}$ | $81.6_{\pm0.6}$ | $75.1_{\pm0.5}$ |
| | Decagon (Zitnik et al., 2018) | $32.9_{\pm1.2}$ | $\underline{53.4}_{\pm0.8}$ | $\underline{43.4}_{\pm0.7}$ | $79.5_{\pm1.2}$ | $83.9_{\pm0.9}$ | $\underline{75.9}_{\pm1.1}$ |
| | KGNN (Lin et al., 2020) | $28.9_{\pm0.9}$ | $48.3_{\pm1.3}$ | $38.6_{\pm1.1}$ | $75.4_{\pm0.9}$ | $74.8_{\pm1.0}$ | $65.2_{\pm0.8}$ |
| | SumGNN (Yu et al., 2021) | $29.1_{\pm0.6}$ | $47.2_{\pm0.9}$ | $36.8_{\pm0.7}$ | $74.0_{\pm0.8}$ | $76.4_{\pm1.0}$ | $70.0_{\pm1.5}$ |
| Text-based | w/ drug id | $21.1_{\pm0.5}$ | $46.9_{\pm0.7}$ | $35.3_{\pm0.6}$ | $67.4_{\pm0.4}$ | $72.9_{\pm0.3}$ | $67.1_{\pm0.5}$ |
| | w/ drug name | $42.1_{\pm0.4}$ | $62.1_{\pm0.3}$ | $54.5_{\pm0.3}$ | $72.3_{\pm0.2}$ | $78.9_{\pm0.3}$ | $75.2_{\pm0.3}$ |
| | w/ truncated descriptions | $48.0_{\pm0.6}$ | $65.8_{\pm0.5}$ | $58.6_{\pm0.4}$ | $84.2_{\pm0.3}$ | $85.3_{\pm0.4}$ | $77.2_{\pm0.6}$ |
| | w/o information selector | $\underline{48.8}_{\pm0.6}$ | $65.2_{\pm0.4}$ | $58.1_{\pm0.3}$ | $\underline{85.3}_{\pm0.3}$ | $\underline{85.6}_{\pm0.4}$ | $78.3_{\pm0.5}$ |
| | TextDDI | $\mathbf{52.5}_{\pm0.7}$ | $\mathbf{67.3}_{\pm0.4}$ | $\mathbf{60.5}_{\pm0.4}$ | $\mathbf{88.2}_{\pm0.2}$ | $\mathbf{88.4}_{\pm0.3}$ | $\mathbf{80.3}_{\pm0.6}$ |
| | (relative improvement) | 19.6% | 13.9% | 17.1% | 8.7% | 4.5% | 4.4% |

drug ids (w/ drug id), a DDI predictor with only drug names (w/ drug name), a DDI predictor with truncated description (w/ truncated description) and a DDI predictor with randomly generated prompts (w/o information selector). For all the variants, task instructions are provided.

### 4.1.3 Metrics

For the DrugBank dataset, which involves a multi-class classification task, we're using three key metrics: macro F1-Score, Accuracy, and Cohen's Kappa (Cohen, 1960). The F1-Score is calculated as the mean of the harmonic mean of precision and recall for each interaction type. Accuracy calculates the correct prediction rate, while Cohen's Kappa assesses the accuracy beyond chance agreement. The TWOSIDES dataset requires multi-label predictions with each interaction type considered as a binary classification problem. Metrics used include ROC-AUC (Fawcett, 2006), PR-AUC (Saito and Rehmsmeier, 2015), and Accuracy. ROC-AUC and PR-AUC gauge the performance of binary classification by analyzing true positives and false positives, and precision and recall respectively. Due to the imbalanced occurrence of DDI types, we chose the F1-Score as primary metric for the Drugbank dataset and PR-AUC for the TWOSIDES dataset because it simultaneously considers precision and recall. More details of the metrics are provided in Appendix C.

### 4.1.4 Implementation details

Unless otherwise specified, for the language model, we utilize RoBERTa-Base (Liu et al., 2019) with a maximum prompt length limit of $L = 256$, along with the Adam optimizer (Kingma and Ba, 2014). The performance of TextDDI with different LMs is provoided in Appendix F. In Algorithm 1, for both the DrugBank and TWOSIDES datasets, the DDI predictor and information selector undergo two training rounds until achieving stable performance on $\mathcal{S}_{\text{val}}$. Each training process independently converges the respective modules. To control the variance, the well-known Proximal Policy Optimization (PPO) (Schulman et al., 2017) algorithm is used to maximize Eq.(6). All experiments are conducted on a server equipped with 8 NVIDIA GeForce RTX 3090 GPUs. More details regarding hyperparameters are provided in Appendix D.

### 4.2 Zero-shot DDI Prediction

From the comparison in Table 2, we have the following notable observations: (i) Among the Pure-DDI methods, MSTE performs worse than MLP. This shortfall is due to MSTE's KG embedding technique which cannot update the embeddings of new drugs, which has no interaction in training, owing to the lack of DDIs involving new drugs.

(ii) Among the KG-based methods, KG-DDI uses the KG embedding technique as well but outperforms MSTE. This superior performance is achieved by updating the embeddings of new

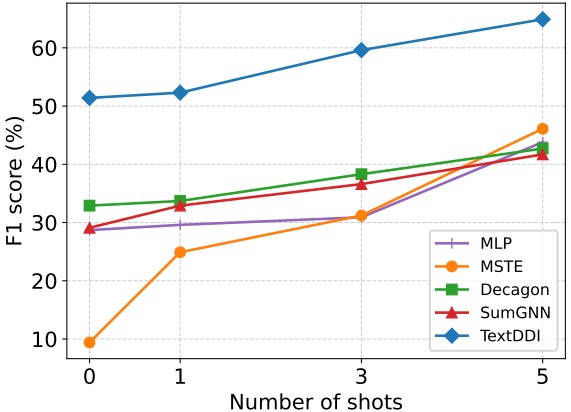

Figure 3: Comparison with baselines under the different number of shots of DDIs seen during training.

drugs using entities in the biomedical KG. Regarding deep GNN-based methods such as CompGCN, Decagon, KGNN, and SumGNN, their performance is remarkably similar. This is mainly attributed to their common use of Graph Neural Network (GNN) architecture for updating drug embeddings, along with their application of connections between the new drug and entities in the biomedical KG to update the embedding of the drug. In general, the deep GNN-based methods outperform the shallow ones like KG-DDI, MSTE, and MLP.

(iii) Among the Text-based methods, it is evident our TextDDI outperforms previous baselines by a significant margin. It is noteworthy that just using the drug name (w/ drug name) already surpasses most of the baselines based on graph methods. We found this is because the naming of drugs generally follows specific patterns. For instance, statin class drugs are used for lowering blood pressure, and drugs like Lovastatin (DB00227) and Fluvastatin (DB01095) have the same suffix. When we convert the drug names to drug IDs (e.g., DB00227), the model's performance (w/ drug id) significantly declines, proving that valuable information is also contained in the drug names. At the same time, we observed that using the truncated drug descriptions as prompts (w/ truncated description) and randomly generated prompts (w/o information selector) to predict DDIs have similar performance. When the information selector is removed, either w/ truncated description or w/o information selector, the model's performance declines, indicating that the information selector module can enhance the DDI predictor's performance by generating more accurate prompts.

## 4.3 Few-shot DDI Prediction

In Section 4.2, we studied the case of no interaction on new drugs. However in practice, there can be some known knowledge on the newly developed drugs. Hence, in this part, we compare the different models' abilities in a few-shot setting, where there can be a few interactions for the new drugs. Specifically, we move 1, 3 and 5 samples (names as 1, 3 and 5 shot, respectively) of each new drug into the training set to setup this scenario.

We compare our method with several typical baselines, including Pure-DDI methods (MLP, MSTE) and KG-based methods (Decagon, SumGNN) on the DrugBank dataset. The results are shown in Figure 3. We can observe that the performance of all methods improves as the number of shots increases. Notably, MSTE exhibits the most significant improvement due to its heavy dependency on known DDIs to predict DDIs for each drug. KG-based approaches such as Decagon and SumGNN exhibit comparable performance growth, attributable to their similar GNN architecture. Once again, in the few-shot setting, our method still outperforms the baselines significantly, as our model can efficiently learn from the DDIs of new drugs.

We also evaluate the performance of TextDDI in the vanilla setting in Yu et al. (2021) without specially caring about the new drugs. The experimental results are provided in the Appendix E.

## 4.4 Case Study on the Selected Information

In this section, we conduct a case study to visualize the prompt generated by the information selector. From the examples presented in Table 3, we observe the huge distinction between the random prompt and the learned prompt. The information in the randomly generated prompt is very diverse and it fails to predict the correct interaction type. On the other hand, the prompt generated by the information selector provides comprehensive descriptions of Aclidinium and Butylscopolamine as "anticholinergic" drugs that act through distinct mechanisms of action. The drug descriptions (with several "anticholinergic" and "muscarinic" related messages) provided by the information selector assist the DDI predictor in accurately predicting the DDI type, i.e., "Aclidinium may increase the anticholinergic activities of Butylscopolamine". This indicates that our method possesses a certain degree of interpretability.

Table 3: Comparison between the randomly generated prompt and the prompt generated by the information selector. Red and green highlight incorrect and correct predictions by the DDI predictor, while orange marks DDI type-related keywords. Refer to Appendix G for more cases.

| Randomly generated prompt | Aclidinium: Aclidinium does not prolong the QTc interval or have significant effects on cardiac rhythm. Aclidinium bromide inhalation powder is indicated for the long-term, maintenance treatment of bronchospasm associated with chronic obstructive pulmonary disease (COPD), including chronic bronchitis and emphysema. It has a much higher propensity to bind to muscarinic receptors than nicotinic receptors. FDA approved on July 24, 2012. Prevention of acetylcholine-induced bronchoconstriction effects was dose-dependent and lasted longer than 24 hours. Butylscopolamine: Used to treat abdominal cramping and pain. Scopolamine butylbromide binds to muscarinic M3 receptors in the gastrointestinal tract. The inhibition of contraction reduces spasms and their related pain during abdominal cramping. In the above context, we can predict that the drug-drug interaction between Aclidinium and Butylscopolamine is that: Aclidinium may decrease the bronchodilatory activities of Butylscopolamine. |
|---|---|
| Prompt generated by the information selector | Aclidinium: Prevention of acetylcholine-induced bronchoconstriction effects was dose-dependent and lasted longer than 24 hours. Aclidinium is a long-acting, competitive, and reversible anticholinergic drug that is specific for the acetylcholine muscarinic receptors. It binds to all 5 muscarinic receptor subtypes to a similar affinity. Aclidinium's effects on the airways are mediated through the M3 receptor at the smooth muscle to cause bronchodilation. Butylscopolamine: This prevents acetylcholine from binding to and activating the receptors which would result in contraction of the smooth muscle. Scopolamine butylbromide binds to muscarinic M3 receptors in the gastrointestinal tract. The inhibition of contraction reduces spasms and their related pain during abdominal cramping. In the above context, we can predict that the drug-drug interaction between Aclidinium and Butylscopolamine is that: Aclidinium may increase the anticholinergic activities of Butylscopolamine. |

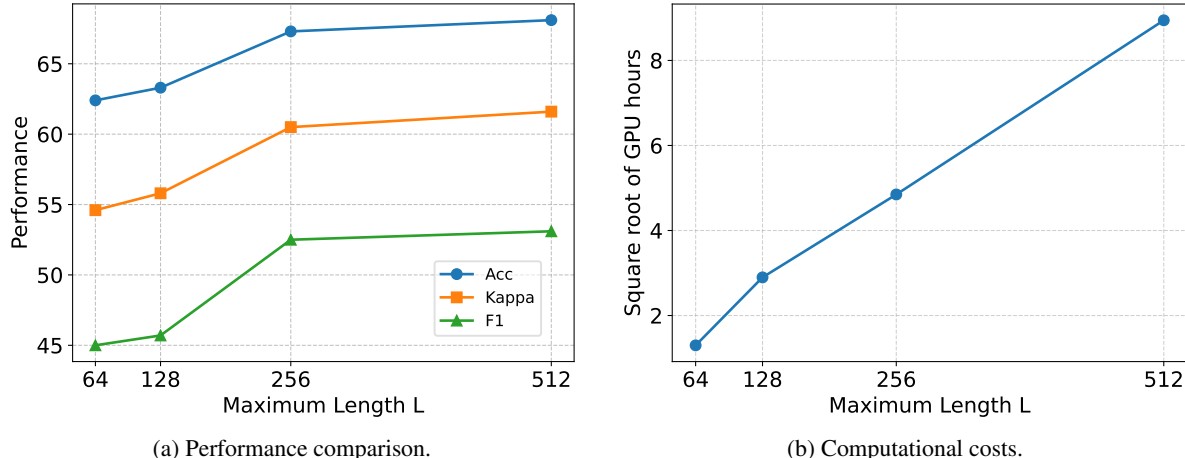

(a) Performance comparison.  (b) Computational costs.

Figure 4: The impact of maximum prompt length $L$ on model performance, and computational costs.

## 4.5 Ablation Study: Length of Prompt

In this section, we conduct ablation experiments on the DrugBank dataset to investigate the impact of prompt length on performance and computational costs. As shown in Figure 4a, when the length of the prompt increases from 64 to 128 and 256, the model's performance significantly improves. However, when the length further increases to 512, the performance improvement becomes negligible, possibly due to a decrease in the density of task-relevant information within the longer prompt. As depicted in Figure 4b, it can be observed that the square root of the GPU hours necessary for achieving model convergence exhibits a linear correlation with the maximum length of the prompt. This finding supports the quadratic association between prompt length and computational costs. Therefore, shortening the length of the prompt is necessary. Thanks to our information selector's ability to generate shorter and more relevant text, we can enhance model performance while conserving computational costs.

## 5 Conclusion

We propose a problem setup that predicts DDI for new drugs as a zero-shot setting. To achieve this, we propose TextDDI, a novel framework that capitalizes on textual descriptions of each single drug. TextDDI consists of an LM-based DDI predictor and an RL-based information selector. The LM-based DDI predictor is trained with drug-pair-dependent descriptions to capture domain-specific knowledge. The RL-based information selector is designed to generate shorter but more relevant descriptions for each drug pair and is trained with DDI predictor serving as the supervisory signal. Experimental results demonstrate that TextDDI achieves better DDI prediction performance than the existing symbolic approaches under zero-shot and few-shot settings. We also show that the descriptions generated by the information selector are semantically relevant to the target prediction.

## Limitations

Our proposed TextDDI for zero-shot DDI prediction still contains some limitations. TextDDI may fail when lacking descriptions of drugs, such as the background, indications, and mechanisms of action. The greedy search used in our information selector can be replaced with beam search strategy, which may lead to better performance. Due to computation resources, the largest model we use is limited to GPT-2 XL (1.5B, in Appendix F). We believe that TextDDI has great potential for improvement in future works.

## Ethics Statement

We follow the ACL Code of Ethics. In our work, there are no human subjects and informed consent is not applicable.

## Acknowledgments

Fangqi Zhu, Bing Qin and Ruifeng Xu's works were partially supported by the National Natural Science Foundation of China 62176076), Shenzhen Foundational Research Funding JCYJ20220818102415032, Guangdong Provincial Key Laboratory of Novel Security Intelligence Technologies 2022B1212010005. Lei Chen's work was partially supported by National Science Foundation of China (U22B2060, 61729201), the Hong Kong RGC GRF Project 16213620, CRF Project C2004-21GF, RIF Project R6020-19, AOE Project AoE/E-603/18, Theme-based project TRS T41-603/20R, Guangdong Basic and Applied Basic Research Foundation 2019B151530001, Hong Kong ITC ITF grants MHX/078/21 and PRP/004/22FX, Microsoft Research Asia Collaborative Research Grant and HKUST-Webank joint research lab grants.

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

## A GPTs For the Zero-shot DDI Prediction

In this section, we evaluate the performance of the state-of-the-art general large language models, including GPT-3.5-Turbo (the API version of Chat-GPT) and GPT4, on the zero-shot DDI prediction task.

The task instruction template we utilized follows the style employed by Gao et al. (2023), which includes task description, interaction types and their definitions, positive example and negative example. Please refer to Table 9 for details about our task instruction template. We use randomly generated prompts to evaluate the performance of the large language models.

Due to the request limitations of GPT4, the evaluated dataset we utilized consists of 100 randomly selected samples from the zero-shot DDI prediction test set $\mathcal{S}_{tst}$ on the DrugBank dataset. For the predictions that cannot be exactly matched with the definition of DDI types, we select the DDI type by calculating the similarity between the prediction text and the definition of DDI types. We also attempt to have GPT-4 use a browser plugin to answer questions. However, in this scenario, GPT-4 was unable to follow instructions and predict DDI types. The results of our evaluation are shown in Table 4.

Table 4: Comparison of TextDDI, GPT-3.5-Turbo, and GPT4 on the zero-shot DDI prediction task. The Miss Rate refers to the proportion of instances that fails to follow the instruction to predict an exact definition of an interaction type.

| Methods | F1-Score | Accuracy | Kappa | Miss Rate |
|---|---|---|---|---|
| Random Guess | 0.0 | 0.0 | -0.4 | - |
| GPT-3.5-Turbo | 0.9 | 2.2 | 1.3 | 7% |
| GPT4 | 6.6 | 10.0 | 8.1 | 4% |
| TextDDI | 35.2 | 55.0 | 45.2 | - |

From Table 4, we observe that the performance of GPT-3.5-Turbo is only marginally better than random guessing, primarily due to its lack of domain knowledge. GPT-4, on the other hand, demonstrates remarkably enhanced capabilities compared to GPT-3.5-Turbo, especially in complex and specialized tasks such as zero-shot DDI prediction. However, neither of both large language models, GPT-3.5-Turbo or GPT-4, can achieve the zero-shot DDI prediction task as effectively as TextDDI, underscoring the significance of our research.

## B Distribution of Entire Prompt Lengths

We present the distribution of prompt lengths for each drug pair using complete drug descriptions in Figure 5. The maximum length for text drug descriptions from websites is 4081. The maximum length for the raw descriptions of drug pairs is 6575. As discussed in Section 4.5, the computational costs increase quadratically with the length of the prompt. Consequently, training a fine-tuning model with prompts consisting of several thousand tokens becomes costly. Therefore, reducing the length of the prompt is necessary, which demonstrates the necessity of our information selector.

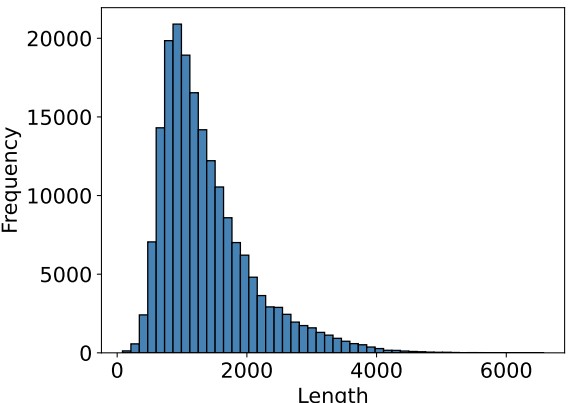

Figure 5: Distribution of prompt length for complete drug-pair prompts constructed with raw drug descriptions.

## C Metric Details

In this section, we provide a more detailed description of the metrics.

For the DrugBank dataset, since there is at most one interaction between a drug pair, the task is a multi-class classification problem. Therefore, we consider the following metrics:

- F1-Score = $\frac{1}{\|\mathcal{R}_D\|} \sum_{r \in \mathcal{R}_D} \frac{2P_r \cdot R_r}{P_r + R_r}$, where $P_r$ and $R_r$ are the precision and recall for the interaction type r, respectively.
- Accuracy: the percentage of accurately predicted interaction type compared with the ground-truth interaction type.
- Cohen's Kappa (Cohen, 1960): $\kappa = \frac{P_o - P_e}{1 - P_e}$, where $P_o$ is the observed agreement (accuracy) and $P_e$ is the chance agreement (the probability of randomly seeing each class).

For the TWOSIDES dataset, since there may be multiple interactions between a drug pair, the task is a multi-label prediction problem, where each

Table 5: Comparison of different methods on the vanilla DDIs prediction task, which predicts interactions between two known drugs. Our results include the average and standard deviation across five distinct random seeds. The best values are in boldface and the second best are underlined.

| Datasets | | DrugBank | | | TWOSIDES | | |
|---|---|---|---|---|---|---|---|
| Type | Methods | F1-Score | Accuracy | Kappa | PR-AUC | ROC-AUC | Accuracy |
| Pure-DDI | MLP (Rogers and Hahn, 2010) | $61.1_{\pm0.4}$ | $82.1_{\pm0.3}$ | $80.5_{\pm0.2}$ | $82.6_{\pm0.3}$ | $81.2_{\pm0.1}$ | $73.5_{\pm0.3}$ |
| | MSTE (Yao et al., 2022) | $83.0_{\pm1.3}$ | $85.4_{\pm0.7}$ | $82.8_{\pm0.8}$ | $90.2_{\pm0.1}$ | $91.3_{\pm0.1}$ | $84.1_{\pm0.1}$ |
| KG-based | KG-DDI (Karim et al., 2019) | $52.2_{\pm1.1}$ | $61.5_{\pm2.8}$ | $55.9_{\pm2.8}$ | $88.2_{\pm0.1}$ | $90.7_{\pm0.1}$ | $83.5_{\pm0.1}$ |
| | CompGCN (Vashishth et al., 2020) | $74.3_{\pm1.2}$ | $78.8_{\pm0.8}$ | $75.0_{\pm1.1}$ | $90.6_{\pm0.3}$ | $92.3_{\pm0.3}$ | $84.8_{\pm0.3}$ |
| | Decagon (Zitnik et al., 2018) | $57.4_{\pm0.3}$ | $87.2_{\pm0.4}$ | $86.1_{\pm0.1}$ | $90.6_{\pm0.3}$ | $91.7_{\pm0.3}$ | $82.1_{\pm0.5}$ |
| | KGNN (Lin et al., 2020) | $74.0_{\pm0.1}$ | $90.9_{\pm0.2}$ | $89.6_{\pm0.2}$ | $90.8_{\pm0.2}$ | $92.8_{\pm0.1}$ | $86.1_{\pm0.1}$ |
| | SumGNN (Yu et al., 2021) | $\underline{86.9}_{\pm0.4}$ | $\underline{92.7}_{\pm0.1}$ | $\underline{90.7}_{\pm0.1}$ | $\underline{93.4}_{\pm0.1}$ | $\underline{94.9}_{\pm0.2}$ | $\underline{88.8}_{\pm0.2}$ |
| Text-based | TextDDI | $\mathbf{91.7}_{\pm0.6}$ | $\mathbf{96.1}_{\pm0.5}$ | $\mathbf{95.4}_{\pm0.4}$ | $\mathbf{94.1}_{\pm0.5}$ | $\mathbf{95.2}_{\pm0.4}$ | $\mathbf{89.0}_{\pm0.6}$ |

type of interaction is treated as a binary classification problem. Following (Zitnik et al., 2018; Tatonetti et al., 2012), we sample one negative drug pair and assess the performance of binary classification with the following metrics:

- ROC-AUC is the area under curve of receiver operating characteristics, measured by $\sum_{k=1}^{n} TP_k \triangle FP_k$ , where $(TP_k, FP_k)$ is the k-th true positive and false positive operating point.
- PR-AUC is the area under curve of precision-recall, measured by $\sum_{k=1}^{n} P_k \triangle R_k$ , where $(P_k, R_k)$ is the k-th precision and recall operating point.
- Accuracy is the percentage of accurately predicted interaction type compared with the ground-truth interaction type.

## D Training Details

We provide more training details in this section. We use generalized Advantage Estimation (GAE) to smooth the rewards. All the hyperparameters are listed in Table 6. For further information on the PPO algorithm, please refer to https://spinningup.openai.com/en/latest/algorithms/ppo.html.

## E Vanilla DDI Prediction

In this section, we conduct experiments using the vanilla setting, which has been commonly employed in previous studies. This setting does not involve any specific handling of new drugs, and it ensures that all drugs are evenly distributed across the training, validation, and test sets. The dataset partitioning described in Yu et al. (2021) is utilized for this purpose. The corresponding results are presented in Table 5. We can observe from Table 5

Table 6: Hyperparameters used for TextDDI across all settings.

| | Hyperparameter Value |
|---|---|
| Learning rate | 0.00001 |
| Weight Decay | 0.000006 |
| Batch Size | 128 |
| Discount Factor $\gamma$ | 0.99 |
| GAE Lambda | 0.95 |
| PPO Epoch | 1 |
| Number of Mini Batch | 4 |
| Clip Ratio | 0.1 |
| Value loss Coefficient | 0.5 |
| Entropy Coefficient | 0.01 |

that TextDDI still achieves a significant lead using the vanilla setting. We also analyze the reason why TextDDI's lead on the TWOSIDES dataset is not as pronounced as on the DrugBank dataset. This may be due to the lower quality of textual information for drugs in the TWOSIDES dataset. Some drugs are named using organic compound nomenclature, such as *3-carboxy-2-hydroxy-N,N,N-trimethylpropan-1-aminium*. In such cases, more data cleaning work is required to replace organic compound names with common drug names.

## F Comparison of Different Language Models

In this section, we evaluate the performance of TextDDI using different language models, including encoder models (RoBERTa-Base and RoBERTa-Large), as well as decoder models scale to different sizes (GPT2, GPT2-Medium, GPT2-Large, and GPT2-XL). For the decoder model, we utilize the output embedding of the final hidden

Table 7: Performance comparison of different language models on the zero-shot DrugBank dataset.

| LM | Parameters Num. | F1-Score | Accuracy | Kappa |
|---|---|---|---|---|
| RoBERTa-Base | 125M | 52.5 | 67.3 | 60.5 |
| RoBERTa-Large | 355M | 51.4 | 66.1 | 58.6 |
| GPT2 | 124M | 47.4 | 65.2 | 57.6 |
| GPT2-Medium | 355M | 48.8 | 65.4 | 57.5 |
| GPT2-Large | 774M | 56.6 | 64.3 | 57.0 |
| GPT-XL | 1500M | 61.4 | 64.8 | 57.0 |

state of the final decoder token "" as the representation of input to finetune the model.

Referring to Table 7, we can observe that larger language models generally yield superior results, with our primary focus being the F1-Score for the DrugBank dataset. For all language models, the hyperparameters are the same as shown in Table 6. Additionally, we observe that the performance of the RoBERTa-Large model is unsatisfactory compared to RoBERTa-Base, indicating a requirement for further hyperparameter optimization. And We find that as the model size increases in the GPT2 series models, the F1-Score significantly improves, while the accuracy slightly decreases. This indicates that model performance outcomes can vary based on the evaluation dimension. Overall, larger models tend to achieve better performance. For example, in terms of F1-Score, GPT-XL exhibits a 14% improvement compared to GPT2.

## G More Case Studies

In this section, we presented more examples as shown in Table 8.

Table 8: Further comparison between the randomly generated prompts and the prompts generated by the information selector. The text highlighted in red and green represents the portions of the DDI predictor's predictions that are incorrect and correct, respectively. The orange portion represents keywords related to the correct DDI type.

| | | |
|---|---|---|
| Case 1 | Randomly generated prompt | Bupivacaine: As an implant, bupivacaine is indicated in adults for placement into the surgical site to produce postsurgical analgesia for up to 24 hours following open inguinal hernia repair. Bupivacaine, in combination with meloxicam, is indicated for postsurgical analgesia in adult patients for up to 72 hours following foot and ankle, small-to-medium open abdominal, and lower extremity total joint arthroplasty surgical procedures. Bupivacaine is often administered by spinal injection prior to total hip arthroplasty. Verapamil: Verapamil is indicated in the treatment of vasopastic (i.e. Prinzmetal's) angina, unstable angina, and chronic stable angina. Verapamil binds to these channels in a voltage- and frequency-dependent manner, meaning affinity is increased 1) as vascular smooth muscle membrane potential is reduced, and 2) with excessive depolarizing stimulus. In the above context, we can predict that the drug-drug interaction between Bupivacaine and Verapamil is that: The risk or severity of adverse effects can be increased when Bupivacaine is combined with Verapamil. |
| | Prompt generated by the information selector | Bupivacaine: Like, bupivacaine is an amide local anesthetic that provides local anesthesia through blockade of nerve impulse generation and conduction. Bupivacaine crosses the neuronal membrane and exerts its anesthetic action through blockade of these channels at the intracellular portion of their pore-forming transmembrane segments. These impulses, also known as action potentials, critically depend on membrane depolarization produced by the influx of sodium ions into the neuron through voltage-gated sodium channels. The block is use-dependent, where repetitive or prolonged depolarization increases sodium channel blockade. Verapamil: Verapamil is known to interact with other targets, including other calcium channels, potassium channels, and adrenergic receptors. Verapamil binds to these channels in a voltage- and frequency-dependent manner, meaning affinity is increased 1) as vascular smooth muscle membrane potential is reduced, and 2) with excessive depolarizing stimulus. N-, P-, Q-, or T-type). In the above context, we can predict that the drug-drug interactions between Bupivacaine and Verapamil is that: The metabolism of Verapamil can be decreased when combined with Bupivacaine. |
| Case 2 | Randomly generated prompt | Penbutolol: Penbutolol is a $\beta$-1, $\beta$-2 (nonselective) adrenergic receptor antagonist. In human studies, however, heart rate decreases have been similar to those seen with propranolol. When $\beta1$ receptors are activated by catecholamines, they stimulate a coupled G protein that leads to the conversion of adenosine triphosphate (ATP) to cyclic adenosine monophosphate (cAMP). The increase in cAMP leads to activation of protein kinase A (PKA), which alters the movement of calcium ions in heart muscle and increases the heart rate. Teriflunomide: Teriflunomide is a pyrimidine synthesis inhibitor with anti-inflammatory and immunomodulatory properties used to treat patients with the relapsing-remitting form of multiple sclerosis. The FDA label states an important warning about the risk of hepatoxicity and teratogenicity for patients using teriflunomide. In the above context, we can predict that the drug-drug interactions between Penbutolol and Teriflunomide is that: The metabolism of Teriflunomide can be decreased when combined with Penbutolol. |
| | Prompt generated by the information selector | Penbutolol: Penbutolol acts on the $\beta$-1 adrenergic receptors in both the heart and the kidney. When $\beta$-1 receptors are activated by catecholamines, they stimulate a coupled G protein that leads to the conversion of adenosine triphosphate (ATP) to cyclic adenosine monophosphate (cAMP). Penbutolol blocks the catecholamine activation of $\beta1$ adrenergic receptors and decreases heart rate, which lowers blood pressure. The increase in cAMP leads to activation of protein kinase A (PKA), which alters the movement of calcium ions in heart muscle and increases the heart rate. Teriflunomide: What is known is that teriflunomide prevents pyrimidine synthesis by inhibiting the mitochondrial enzyme dihydroorotate dehydrogenase, and this may be involved in its immunomodulatory effect in MS. The exact mechanism by which teriflunomide acts in MS is not known. In the above context, we can predict that the drug-drug interactions between Penbutolol and Teriflunomide is that: Penbutolol may decrease the antihypertensive activities of Teriflunomide. |

Table 9: The task instruction template we utilze, where the **bold** text represents the part that needs to be predicted by the large language model. Please note that the instruction is drug-pair-dependent, and the names $u^{name}$ and $v^{name}$, as well as the descriptions $u^{sel}$ and $v^{sel}$, of the drug pair $(u, v)$ will be filled into the template during the prediction process.

Task Instruction

Task Description: This is a drug-drug interactions prediction task where the goal is to predict the interaction type given a pair of drugs.

Interaction types and their definitions:

#Drug1 may increase the photosensizing activities of #Drug2.
#Drug1 may increase the anticholinergic activities of #Drug2.
The bioavailability of #Drug2 can be decreased when combined with #Drug1.
. . .

Positive Example:
Input: #Drug1: Riboflavin, #Drug2: Verteporfin.
Context:
Riboflavin: The antioxidant activity of riboflavin is principally derived from its role as a precursor of FAD and the role of this cofactor in the production of the antioxidant reduced glutathione. Reduced glutathione is the cofactor of the selenium-containing glutathione peroxidases among other things. The glutathione peroxidases are major antioxidant enzymes. Reduced glutathione is generated by the FAD-containing enzyme glutathione reductase. Binds to riboflavin hydrogenase, riboflavin kinase, and riboflavin synthase.
Verteporfin: Verteporfin appears to somewhat preferentially accumulate in neovasculature, including choroidal neovasculature. Verteporfin is transported in the plasma primarily by lipoproteins. However, animal models indicate that the drug is also present in the retina. As singlet oxygen and reactive oxygen radicals are cytotoxic, Verteporfin can also be used to destroy tumor cells.
Output: #Drug1 may increase the photosensizing activities of #Drug2.

Negative Example:
Input: #Drug1: Riboflavin, #Drug2: Verteporfin.
Context:
Riboflavin: The antioxidant activity of riboflavin is principally derived from its role as a precursor of FAD and the role of this cofactor in the production of the antioxidant reduced glutathione. Reduced glutathione is the cofactor of the selenium-containing glutathione peroxidases among other things. The glutathione peroxidases are major antioxidant enzymes. Reduced glutathione is generated by the FAD-containing enzyme glutathione reductase. Binds to riboflavin hydrogenase, riboflavin kinase, and riboflavin synthase.
Verteporfin: Verteporfin appears to somewhat preferentially accumulate in neovasculature, including choroidal neovasculature. Verteporfin is transported in the plasma primarily by lipoproteins. However, animal models indicate that the drug is also present in the retina. As singlet oxygen and reactive oxygen radicals are cytotoxic, Verteporfin can also be used to destroy tumor cells.
Output: #Drug1 may decrease the vasoconstricting activities of #Drug2.

Evaluation Instance:
Input: #Drug1: $\{u^{name}\}$, #Drug2: $\{v^{name}\}$.
Context:
$\{u^{name}\}$: $\{u^{sel}\}$
$\{v^{name}\}$: $\{v^{sel}\}$
Qusetion: What is your guess for the interaction type between #Drug1: $\{u^{name}\}$, #Drug2: $\{v^{name}\}$, answer one of the above interaction types. It does not have to be fully correct.
Output: **The risk or severity of heart failure can be increased when #Drug2 is combined with #Drug1.**