# OpenReview forum: "Learning to Describe for Predicting Zero-shot Drug-Drug Interactions"
_EMNLP/2023/Conference — EMNLP 2023 Main_

### Official Review · Reviewer_mJfs · 2023-07-31

**Soundness:** 3

**Excitement:**

4: Strong: This paper deepens the understanding of some phenomenon or lowers the barriers to an existing research direction.

**Paper Topic And Main Contributions:**

In this paper, the authors focus on zero-shot drug-drug interaction (DDI) prediction for new drugs. A new approach TextDDI is proposed to leverage textual information. The proposed model consists of a large language model (LLM)-based DDI predictor and a reinforcement learning (RL)-based information selector. The DDI prediction uses a LLM model to capture the domain knowledge, and then the information selector utilizes RL to describe the drugs with shorter but more relevant text for each drug pair. The experimental results on the DrugBank and TWOSIDES datasets show that the proposed model outperforms all baselines.

**Questions For The Authors:**

1. What is the maximum length of the text drug descriptions from websites? What is the proportion of texts that exceed the model's maximum input length?
2. The results of three metrics (i.e., F1-scores, Accuracy and Kappa) show inconsistent trends. Which metrics do you prefer to be the main metric?

**Reasons To Accept:**

1. The authors propose a novel problem setup that leverages textual descriptions for zero-shot DDI prediction for new drugs.
2. The proposed TextDDI applies the LLM for the DDI predictor and reinforcement learning to further improve the performance. The experimental results on two datasets show that the proposed model outperforms all baselines.

**Reasons To Reject:**

1. The reason why the proposed method outperforms other baselines is unclear. From Table 2, the proposed method without descriptions also achieves promising results on two datasets and even outperforms all baselines on the DrugBank dataset. The proposed method without descriptions means that only the text of drug names is used for DDI predictions. Why can the model using only drug name text outperform other baselines using more drug information? The authors should analyze and explain this.
2. The proposed DDI predictor using all textural information (Truncate if the text is over max length) should be added for comparison.

**Reproducibility:**

4: Could mostly reproduce the results, but there may be some variation because of sample variance or minor variations in their interpretation of the protocol or method.

**Reviewer Confidence:**

4: Quite sure. I tried to check the important points carefully. It's unlikely, though conceivable, that I missed something that should affect my ratings.

**Typos Grammar Style And Presentation Improvements:**

The description of algorithm 1 is not standardized. The algorithm should have inputs and outputs.

---

> ### Author Rebuttal · Authors · 2023-08-29
>
> **Q1**: *The reason why the proposed method outperforms other baselines is unclear. From Table 2, the proposed method without descriptions also achieves promising results on two datasets and even outperforms all baselines on the DrugBank dataset. The proposed method without descriptions means that only the text of drug names is used for DDI predictions. Why can the model using only drug name text outperform other baselines using more drug information? The authors should analyze and explain this.*
>
> **A1**:
> Thank you for your reminder.
>
> During the experimental process, we have considered why the performance of the DDI predictor using only drug names almost outperforms all baselines. In the supplementary materials, our submitted code's configuration file contains a `use_id` parameter, indicating that only the drug's ID, such as DB00001 is used for prediction as a futher ablation for drug names(*w/o description*). The experimental results for drug id only are as follows:
>
> For the DrugBank dataset:
>
> | Method         | F1-Score | Accuracy | Kappa |
> | -------------- | -------- | -------- | ----- |
> | drug id only   | 21.1     | 46.9     | 35.3  |
> | drug name only | 42.1     | 62.1     | 54.5  |
> | TextDDI        | 52.5     | 67.3     | 60.5  |
>
> For the TWOSIDES dataset:
>
> | Method                     | PR-AUC   | ROC-AUC  | Accuracy |
> |----------------------------|----------|----------|----------|
> | drug id only               | 67.4     | 72.9     | 67.1     |
> | drug name only             | 72.3     | 78.9     | 75.2     |
> | TextDDI                    | 88.2     | 88.4     | 80.3     |
>
> We can observe that, when drug names are replaced with drug IDs without using other descriptions, the performance of the DDI predictor significantly decreases, performing only slightly better than MSTE among the seven baselines on the both datasets.
>
> The experimental results demonstrate that **drug names may also contain valuable information**, as the nomenclature of drugs typically follows certain conventions. These names are often simple and descriptive, making it easier for professionals to identify the category and function of the drugs. For instance, statin drugs like Lovastatin (DB00227) and Fluvastatin (DB01095) are both used to lower cholesterol and prevent cardiovascular diseases. This similarity could assist models in predicting on unseen statin drugs such as Simvastatin (DB00641) and Atorvastatin (DB01076).
> Another illustrative example of informative drug naming involves proton pump inhibitors like Omeprazole (DB00338), Lansoprazole (DB00448), and Pantoprazole (DB00213). They are all used to reduce gastric acid secretion and treat gastric ulcers, and they share the common ending "-prazole."
>
> We have also found some examples of similar DDIs (drug-drug interactions) for Lovastatin and Simvastatin with the same drugs, such as Abatacept and Acalabrutinib:
> - The metabolism of Lovastatin can be increased when combined with Abatacept.
> - The metabolism of Simvastatin can be increased when combined with Abatacept.
> - The metabolism of Lovastatin can be decreased when combined with Acalabrutinib.
> - The metabolism of Simvastatin can be decreased when combined with Acalabrutinib.
>
> **These examples can help us explain how the model makes zero-shot DDI predictions solely based on drug names.**
>
> In summary, we attribute the strong performance of the DDI predictor that only uses drug names for prediction(*w/o description*) to: 1. Drug names being highly condensed representations of drug information and 2. The powerful capabilities of pre-trained models. We will include this analysis in Section 4.2 *Zero-shot DDI Prediction* of future versions.
>
>
>
>
> **Q2**: *The proposed DDI predictor using all textural information (Truncate if the text is over max length) should be added for comparison.*
>
> **A2**: Thank you for your suggestion again. We share the same idea, and we have actually implemented this baseline (textDDI with truncated description) before.
> In the supplementary materials, our submitted code's configuration file contains a `not_random` parameter, which corresponds to the implementation of the truncated description baseline.
>
> **However, this baseline performs similarly to predicting with randomly selected sentences (*w/o information selector*), as shown in the table below. Hence, we did not discuss it in the current version considering the space constraints.** We will discuss this baseline in future versions.
>
> For the DrugBank dataset:
>
> | Method                     | F1-Score | Accuracy | Kappa |
> |----------------------------|----------|----------|-------|
> | truncated description      | 48.0     | 65.8     | 58.6  |
> | w/o information selector   | 48.8     | 65.2     | 58.1  |
>
> For the TWOSIDES dataset:
>
> | Method                     | PR-AUC   | ROC-AUC  | Accuracy |
> |----------------------------|----------|----------|----------|
> | truncated description      | 84.2     | 85.3     | 77.2     |
> | w/o information selector   | 85.3     | 85.6     | 78.3     |
>
> **Q3**: *What is the maximum length of the text drug descriptions from websites? What is the proportion of texts that exceed the model's maximum input length?*
>
> **A3**:
>
> The maximum length for text drug descriptions from websites is 4081.
> The maximum length for the raw descriptions of drug pairs is 6575, as indicated in Figure 5 of Appendix B.
>
> Appendix B displays the length distribution of drug descriptions(for drug pairs). Among them
>
> - 99% of drug pairs have raw drug description lengths exceeding our set maximum input length of 256.
> - 98% of drug pairs have raw drug description lengths exceeding the maximum length of the RoBERTa model, which is 512.
> - 63% of drug pairs have raw drug description lengths exceeding the maximum length of the GPT-2 model, which is 1024.
>
> For the description of a single drug:
>
> - 85% of drugs have raw drug description lengths exceeding our set maximum input length of 256.
> - 42% of drugs have raw drug description lengths exceeding the maximum length of the RoBERTa model, which is 512.
> - 12% of drugs have raw drug description lengths exceeding the maximum length of the GPT-2 model, which is 1024.
>
> We will add the above information in Appendix B in future versions.
>
> **Q4**: *The results of three metrics (i.e., F1-scores, Accuracy and Kappa) show inconsistent trends. Which metrics do you prefer to be the main metric?*
>
> **A4**: **In #line 910, we mentioned that we adopted the F1-Score as our main metric.** We chose the F1-Score because it assigns equal weight to each DDI type. Due to the imbalanced occurrence of DDIs, for instance, the most frequently occurring DDI type among 86 DDI types in the DrugBank dataset accounts for nearly 29%, using F1-Score as the main evaluation metric is fairer.
>
> To further investigate this inconsistency trend, we assigned numbers to ten models as follows: MLP (0), MSTE (1), KG_DDI (2), CompGCN (3), Decagon (4), KGNN (5), SumGNN (6), w/o description (7), w/o information selector (8), and TextDDI (9). The rankings of these models based on three metrics (from lowest to highest) are as follows:
>
> F1-Score: 1 2 0 5 6 3 4 7 8 9
>
> Accuracy: 0 1 6 5 2 3 4 7 8 9
>
> Kappa   : 0 1 6 2 5 3 4 7 8 9
>
> We observed that the inconsistency between metrics decreases when the model performance is better, and the top-performing five models exhibit the same trend (i.e., 3 4 7 8 9). Additionally, we found that the trend in Accuracy closely aligns with Kappa, possibly due to Kappa being derived from Accuracy.
>
> **For the TWOSIEDS dataset, for the same reasons, we chose PR-AUC as the main metric because it simultaneously considers precision and recall.**
>
> We will clarify this in future versions.
>
>
>
>
> **Q5**: *The description of algorithm 1 is not standardized. The algorithm should have inputs and outputs.*
>
> **A5**: Thank you for your suggestion. In Algorithm 1, "`Require`" indicates the input of the algorithm, which includes the DDI predictor, the information selector before training, the training set $S_{tra}$ , and the validation set $S_{val}$ . TextDDI iteratively performs supervised fine-tuning on the DDI predictor and uses reinforcement learning to train the information selector, supervised by the DDI predictor. In the end, "`return`" term represents the output of the algorithm, including DDI predictor and information selector after training. In future versions, we will standardize the algorithm by replacing "require" with "inputs" and adding "outputs" row below, while removing the "return" row.

---

### Official Review · Reviewer_Na97 · 2023-08-03

**Soundness:** 4

**Excitement:**

4: Strong: This paper deepens the understanding of some phenomenon or lowers the barriers to an existing research direction.

**Missing References:**

There is substantial research on using reinforcement learning for text selection beyond prompt learning. For example, substantial research has been conducted using reinforcement learning for summarization and paraphrasing. Expanding on the related work section in this area could be useful.


REFERENCES:
Yin, Haiyan, Dingcheng Li, and Ping Li. "Learning to selectively learn for weakly supervised paraphrase generation with model-based reinforcement learning." Proceedings of the 2022 Conference of the North American Chapter of the Association for Computational Linguistics: Human Language Technologies. 2022.

Lee, Gyoung Ho, and Kong Joo Lee. "Automatic text summarization using reinforcement learning with embedding features." Proceedings of the Eighth International Joint Conference on Natural Language Processing (Volume 2: Short Papers). 2017.

**Paper Topic And Main Contributions:**

This paper introduces a novel method of predicting drug-drug interactions (DDI) using drug descriptions with a particular emphasis on zero-shot prediction (i.e., drugs that have not been seen in the training dataset). The method works using two components: a DDI predictor and an information selector. The DDI predictor is trained using random snippets from the descriptions of two drugs as input. The information selector uses reinforcement learning to identify the sentences in the description that maximizes model performance. The model performance of the proposed method substantially outperforms prior work.

**Questions For The Authors:**

Question A: Would the performance be as high as reported for truly new drugs? When would this method fail?

**Reasons To Accept:**

Overall, the experiments in this paper are very well done, with averages reported across random seeds, detailed descriptions of the training, development, and test datasets, computational cost experiments, and a useful case study depicting why the method works.

I really liked the idea of the information selector. The approach could be relevant beyond the descriptions in DrugBank, e.g., PubMed research articles. Ultimately, the approach seems generalizable.

**Reasons To Reject:**


(Minor) The search for relevant information seems greedy. I imagine there are cases where the greedy addition of information can go haywire, resulting in incorrect predictions. It would be useful to see cases when this happens. I assume using beam search or some other method of optimizing the sequence at inference time could help, but it may not completely mitigate the issue.

(Minor) This approach assumes a large description of the new Drug in a very specific style. However, for real "new" drugs, this description may be unavailable or unreliable/incomplete. The current evaluation pipeline assumes that these new descriptions are easily accessible, available, and informative. So, the zero-shot results may be overestimated in practice. A solution to this "cold-start" issue may be required for accurate performance. I think the limitations section could be expanded on these additional points. Note: This is probably an issue with prior work as well.

(Very Minor) The use of kappa scores seems a bit out-of-place. Instead of using kappa scores o evaluate if the method is better than random, the proposed could be compared to random, and similar, baselines (e.g., predicting at random, predicting the most common class all the time, etc.), which would be a bit more informative.


**Reproducibility:**

4: Could mostly reproduce the results, but there may be some variation because of sample variance or minor variations in their interpretation of the protocol or method.

**Reviewer Confidence:**

4: Quite sure. I tried to check the important points carefully. It's unlikely, though conceivable, that I missed something that should affect my ratings.

---

> ### Author Rebuttal · Authors · 2023-08-29
>
> **Q1**: *(Minor) The search for relevant information seems greedy. I imagine there are cases where the greedy addition of information can go haywire, resulting in incorrect predictions. It would be useful to see cases when this happens. I assume using beam search or some other method of optimizing the sequence at inference time could help, but it may not completely mitigate the issue.*
>
> **A1**:
> Thank you for your feedback. In general, the advantage of greedy search over beam search is that it has **lower computational overhead** and is **simpler to implement**. However, the downside of greedy search is that it tends to get stuck in local optima and generates sequences that lack diversity. For example, when using greedy search for natural language generation, it often produces repetitive words. TextDDI generates short but more relevant drug descriptions at the sentence level based on the raw description of drug pairs, but the selection of sentences is done without replacement (#line 312 ~ 316), which *avoids the problem of generating duplicate sentences when using greedy search*. Compared to greedy search, beam search has higher complexity because it requires interactions with the DDI predictor every time to obtain the probability of a sentence based on Eq. (4).
>
> Since our method's primary objective is to select the most informative and relevant information, maintaining sentence coherence through beam search is not as crucial. Furthermore, the search strategy is not our main contribution and greedy search has already achieved good performance.
>
> We appreciate your suggestion of taking beam search into consideration. In the conclusion section, we will mention that beam search or its variant strategies could be considered for future work.
>
>
>
> **Q2**: *(Minor) This approach assumes a large description of the new Drug in a very specific style. However, for real "new" drugs, this description may be unavailable or unreliable/incomplete. The current evaluation pipeline assumes that these new descriptions are easily accessible, available, and informative. So, the zero-shot results may be overestimated in practice. A solution to this "cold-start" issue may be required for accurate performance. I think the limitations section could be expanded on these additional points. Note: This is probably an issue with prior work as well.*
>
> **A2**: **Drug-drug interactions (DDIs) are typically assessed during the latter stages of drug development.** The process of developing a new drug generally comprises several phases, such as basic research, preclinical studies, and clinical trials. The information required for TextDDI is primarily provided during the basic research and preclinical studies stages, including the drug's background, indications, and mechanism of action. Clinical trials, on the other hand, primarily focus on the safety and efficacy of the drug, including drug-drug interactions. **Therefore, when predicting drug-drug interactions for a new drug, it is highly likely that researchers already have access to the relevant drug descriptions.**
>
> It's worth noting that many of the baseline methods require Drug-Drug Interaction (DDI) information between the new drug and other drugs, or the linkage of new drugs to the Biomedical Knowledge Graph, in order to understand new drugs. Obtaining such information is more challenging than acquiring the drug descriptions used in our method.
>
> Following your suggestion, we will add the following additional limitation regarding TextDDI in future versions:
> - TextDDI requires drug background, indications, and mechanism of action as descriptive information to help the model understand new drugs, which might limit its application for really new drugs in the early stages of drug development.
>
>
>
>
> **Q3**: *(Very Minor) The use of **kappa** scores seem a bit out-of-place. Instead of using kappa scores to evaluate if the method is better than random, the proposed could be compared to random, and similar, baselines (e.g., predicting at random, predicting the most common class all the time, etc.), which would be a bit more informative.*
>
> **A3**: We consider using the Kappa metric for two reasons:
>
> 1. Kappa is a classic evaluation metric, particularly useful when dealing with imbalanced classification categories like the DDIs in our problem.
> 2. As previous baselines like SumGNN used the kappa metric, we also include kappa as a fair and more comprehensive comparison for different methods.
>
> The concept behind the kappa metric is to measure the difference between the accuracy of a classifier and the accuracy it would achieve through random guessing:
>
> $\kappa = \frac {{P_0 - P_E}} {{1 - P_E}} $
>
> where $P_0$ represents the classifier's accuracy, and $P_E$ describes the accuracy of the classifier making random guesses.
>
> Examining the performance difference between a classifier and one that always predicts the most common class is quite interesting. Similarly, we can define a new kappa as follows:
>
> $\kappa' = \frac {{P_0 - P_C}} {{1 - P_C}} $
>
> where $P_0$ is the classifier's accuracy, and $P_C$ describes the accuracy of a classifier that predicts the most common class.
>
> Taking into account $P_C$=0.290, we can calculate the new kappa metric performance (%) for the 10 methods in Table 2 as follows:
>
> 22.4(MLP), 23.2(MSTE), 27.9(KG-DDI), 29.3(CompGCN), 34.4(Decagon), 27.2(KGNN), 25.6(SumGNN), 46.6(w/o description), 51.0(w/o information selector), 53.9(TextDDI)
>
> We found that the new kappa, $\kappa'$, is linearly correlated with accuracy. Kappa and $\kappa'$ have maintained almost the same trend, except for KG-DDI and CompGCN. Additionally, all baseline models outperform predictions of the most common class (Accuracy=29%). We will consider adding this metric in future versions as you suggested.
>
>
>
> **Q4**: *Would the performance be as high as reported for truly new drugs? When would this method fail?*
>
> **A4**: The years of the drugs in the test set we used range from 2021 to 2022, while the drugs in the validation set span from 2017 to 2020. The training set covers the years from 1907 to 2017. **Hence, the experiment setup in our paper is trying to resemble the case of truly new drugs.**
>
> In practice, we are only available to some public dataset and the truly new drugs may only be known by the drug developers. **Hence, we may not be able to conduct this kind of experiments.**
>
> Since our method has not been applied in practice yet, its performance in predicting Drug-Drug Interactions (DDIs) with truly new drugs is currently uncertain. **One of the limitations of our TextDDI is that it will fail without descriptions of the background, indications, and mechanisms of action of new drugs. (When would this method fail?)**
>
> However, as discussed in A2, we consider safety verification, which includes DDI, to be a later stage in the development of new drugs. Therefore, we believe that the necessary drug description will be obtainable at this stage. Previous methods required harder-to-obtain information for predictions, such as DDIs with other drugs for new drugs, and manual updates to the biomedical knowledge graph for new drugs. In comparison, our method is more amenable to practical application.
>
>
>
> **Q5**: *Missing References: We acknowledge that there is substantial research on using reinforcement learning for text selection beyond prompt learning. For example, substantial research has been conducted using reinforcement learning for summarization and paraphrasing. Expanding on the related work section in this area could be useful.*
>
> REFERENCES: Yin, Haiyan, Dingcheng Li, and Ping Li. "Learning to selectively learn for weakly supervised paraphrase generation with model-based reinforcement learning." Proceedings of the 2022 Conference of the North American Chapter of the Association for Computational Linguistics: Human Language Technologies. 2022.
>
> Lee, Gyoung Ho, and Kong Joo Lee. "Automatic text summarization using reinforcement learning with embedding features." Proceedings of the Eighth International Joint Conference on Natural Language Processing (Volume 2: Short Papers). 2017.*
>
> **A5**: Thank you for your suggestion. We will add a paragraph on **reinforcement learning for text selection** in the *Background* section in future versions. The discussions are provided as follows:
>
> **Reinforcement learning for text selection**
> The prompt generating of drug pairs can be considered as a kind of text selection problem. Within this category, Yin et al. (2022) employed reinforcement learning for weakly-supervised paraphrase generation. They used reinforcement learning to select valuable samples from a large set of weakly labeled sentence pairs for fine-tuning the paraphrase generation pre-trained language models. Lee et al. (2017) utilized reinforcement learning to select sentences for text summarization tasks, using ROUGE-2 score as the reinforcement learning's supervisory signal. However, both of them are unable to address the task of generating sentence sequences related to drug-drug interaction (DDI) prediction for drug pairs in the absence of existing supervision signals.

---

### Official Review · Reviewer_yZ3w · 2023-08-04

**Soundness:** 3

**Excitement:**

4: Strong: This paper deepens the understanding of some phenomenon or lowers the barriers to an existing research direction.

**Paper Topic And Main Contributions:**

This paper addresses the critical challenge in healthcare regarding the prediction of adverse drug-drug interactions (DDIs) through utilizing pre-trained language models and textual knowledge.

Main Contributions:
1. The authors present an innovative approach named TextDDI, which leverages textual information from online databases like DrugBank and PubChem to predict potential drug-drug interactions.

2. The authors propose a novel Reinforcement Learning (RL)-based Information Selector to select the most concise and relevant text for improving the performance of DDI prediction.

3. The authors have made the code and data used in this study available at the provided URL, contributing to the research community's ability to replicate and extend the study.

**Questions For The Authors:**

Q1. From where do you obtain the drug description, and do you verify whether there is a data leakage problem?
Q2. Why is the input length set so short? Is there a compelling reason for this, or is it merely for the convenience of the experiment?

**Reasons To Accept:**

1. The authors propose an innovative approach. The TextDDI approach, which combines Language Models with Reinforcement Learning (RL) for information selection, is a novel and creative solution to the problem.

2. Good Relevance to Real-world Challenges. Adverse DDIs are a critical concern in healthcare, and the ability to predict such interactions, especially for new drugs, has immediate practical implications. Thus, the research addresses a timely and impactful real-world issue.

**Reasons To Reject:**

1. The construction of the dataset is unclear, and it's highly likely that there's a data leakage problem. It is reasonable to include the textual description of known drugs, but this description likely contains information related to drug-drug interaction outcomes. For instance, on Wikipedia, there's a specialized interaction section that discusses potential interactions with other chemicals. See Q1.

2. The improvement brought about by the RL-based information selector may be attributed to the relatively short input length limitation. The authors have set a limit for the input length to 256 tokens, meaning that each drug description can only be 128 tokens long, roughly amounting to 100 words. I believe this is insufficient to encapsulate all necessary information about a drug, making the selection of information valuable. However, if more input is permitted, would the RL selector contribute equally? Especially in today's context, most of the state-of-the-art (SOTA) Large Language Models (LLMs) support input with more than a thousand tokens. I am unsure if this setting is reasonable. See Q2.

**Reproducibility:**

5: Could easily reproduce the results.

**Reviewer Confidence:**

4: Quite sure. I tried to check the important points carefully. It's unlikely, though conceivable, that I missed something that should affect my ratings.

**Typos Grammar Style And Presentation Improvements:**

S1. Most language models the authors utilize are far smaller than the current so-called LLMs. I would suggest not using the term large language models.

---

> ### Author Rebuttal · Authors · 2023-08-29
>
> **Q1**: *The construction of the dataset is unclear.*
>
> **A1**: Thank you for your suggestions.
>
> In #line 388 ~ 391, we introduced that we are using standard datasets, DrugBank and TWOSIDES, following Zitnik et al. (2018) and Yu et al. (2021). In #line 156 ~ 168, we described the source of our drug description information. And Section 4.4.1 *Datasets split for zero-shot DDI* details how we partitioned the dataset for zero-shot DDI.
>
> We will organize these information and incorporate additional information in Section 4.4.1 *Datasets split for zero-shot DDI* in future versions as follows:
>
> - For the DrugBank dataset, Ryu et al. (2018) downloaded 192,303 DDIs described in the form of sentences from the [DrugBank database](https://go.drugbank.com/releases) (version 5.0.3), **Drug Interactions** Section. DDI types that occurred less than 5 times were filtered out, resulting in a total of 86 DDI types and 185, 164 DDIs. For each drug, we utilized information from the [DrugBank website](https://go.drugbank.com/drugs/DB00001) concerning the drugs' *Backgrounds*, *Indications*, and *Mechanisms of action* as the original drug descriptions.
> - For the TWOSIDE dataset, Tatonetti et al. (2012) collected the dataset from large collections of adverse event reports, considering only high-confidence DDIs. For each drug, we use the drug *Description* section on the [PubChem website]((https://pubchem.ncbi.nlm.nih.gov/compound/5206)) as the original description of the drug.
>
> [1] Marinka Zitnik, Monica Agrawal, and Jure Leskovec. 2018. Modeling polypharmacy side effects with graph convolutional networks. Bioinformatics, 34(13):i457–i466.
>
> [2] Yue Yu, Kexin Huang, Chao Zhang, Lucas M Glass, Jimeng Sun, and Cao Xiao. 2021. Sumgnn: multi-typed drug interaction prediction via efficient knowledge graph summarization. Bioinformatics, 37(18):2988–2995.
>
> [3] Ryu,J.Y. et al. (2018) Deep learning improves prediction of drug–drug and drug–food interactions. Proc. Natl. Acad. Sci. USA, 115, E4304–4311.
>
> [4] Tatonetti,N.P. et al. (2012) Data-driven prediction of drug effects and interactions. Sci. Transl. Med., 4, 125ra31.
>
>
>
> **Q2**: *It's highly likely that there's a **data leakage problem**. It is reasonable to include the textual description of known drugs, but this description likely contains information related to drug-drug interaction outcomes. For instance, on Wikipedia, there's a specialized **interaction section** that discusses potential interactions with other chemicals. **Question**: From where do you obtain the drug description, and do you verify whether there is a data leakage problem?*
>
> **A2**: Thank you for your reminder, but **we can confirm that there is no data leakage problem** here because we have already taken this factor into account when obtaining drug descriptions.
>
> -  For the DrugBank dataset, the drug descriptions we utilized include information from the *Backgrounds*, *Indications*, and *Mechanism of action sections* (#line 076 ~ 077) on the [DrugBank website](https://go.drugbank.com/drugs/DB00001), **excluding the Drug Interaction section**. The Backgrounds, Indications, and Mechanism of action sections primarily provide information about the properties of the indexed drugs without including interactions with other drugs.
>
> -  For the TWOSIDES dataset, the drug descriptions we used are from the *Description* section of each drug on the [PubChem website](https://pubchem.ncbi.nlm.nih.gov/compound/5206) , which provides basic information about the indexed drugs. Similarly, there is a separate *Interactions* section on the PubChem website, which we **did not use**.
>
> Reviewers can browse our provided [DrugBank website](https://go.drugbank.com/drugs/DB00001) and [PubChem website](https://pubchem.ncbi.nlm.nih.gov/compound/5206) to further ensure that there are no data leakage issues here.
>
> Thank you for your concern regarding this issue. We will include descriptions to highlight the exclusion of data leakage problem in future versions.
>
>
>
> **Q3**: *The improvement brought about by the RL-based information selector may be attributed to the relatively short input length limitation. The authors have set a limit for the input length to 256 tokens, meaning that each drug description can only be 128 tokens long, roughly amounting to 100 words. I believe this is insufficient to encapsulate all necessary information about a drug, making the selection of information valuable. However, if more input is permitted, would the RL selector contribute equally? Especially in today's context, most of the state-of-the-art (SOTA) Large Language Models (LLMs) support input with more than a thousand tokens. I am unsure if this setting is reasonable. **Question**: Why is the input length set so short? Is there a compelling reason for this, or is it merely for the convenience of the experiment?*
>
> **A3**:
> In regard to why the input length is set to 256, our explanation is as follows:
>
> In Section 4.5 *Ablation Study: Length of Prompt*, we analyzed the effect of input length on both the model's performance and its computational costs. Specifically, in #line 562 ~ 565, we stated："However, when the length further increases to 512, the performance improvement becomes negligible, possibly due to a decrease in the density of task-relevant information within the longer prompt".
>
> In fact, the input length of 256 was determined through hyperparameter tuning. Since we were using the pretrained language model RoBERTa with a maximum input length of 512, we experimented with setting the input length to 64, 128, 256, and 512, as outlined in Figure 4. Ultimately, we found that when training with a length of 512, the model's performance only marginally improved upon 256, while the computational cost increased quadratically. On the DrugBank dataset for example, the performance of F1-score 52.5 at $L=256$ required 23 GPU hours, whereas the F1-score 52.9 at $L=512$ required 80 GPU hours.
>
> **Due to limited computing resources (only 8 NVIDIA GeForce RTX 3090 GPUs) and to reduce the cost of subsequent experiments, we chose a maximum length of 256, with only a slight loss in performance but still outperforming all the baselines.** For fair comparation, we choose $L=256$ to conduct the various subsequent experiments, including the *Few-shot settings* (Section 4.3), *vanilla DDI prediction experiments* (Appendix E), and *Comparison of Different Language Models experiments* (Appendix F). Especially, the GPT-XL experiment required 312 GPU hours at a maximum length of 256, thus setting the maximum length to 512 in this case would be hard to afford, estimated to require more than one week for training.
>
> Regarding the question of whether the information selector can make the same contribution on longer input lengths, we present the ablation experiments of TextDDI on the information selector when the input length is 512 as follows:
>
> For the DrugBank dataset:
>
> | Method                                         | F1-Score | Accuracy | Kappa |
> |------------------------------------------------|----------|----------|-------|
> | w/o information selector (input length=256)    | 48.8     | 65.2     | 58.1  |
> | TextDDI (input length=256)                     | 52.5 (+3.7)     | 67.3 (+2.1)     | 60.5 (+2.4)  |
> | w/o information selector (input length=512)    | 49.3     | 65.5     | 59.4  |
> | TextDDI (input length=512)                     | 52.9 (+3.6)    | 67.9 (+2.4)    | 61.4 (+2.0) |
>
> For the TWOSIDES dataset:
> | Method                                         | PR-AUC | ROC-AUC | Accuracy |
> |------------------------------------------------|----------|----------|-------|
> | w/o information selector (input length=256)    | 85.3     | 85.6     | 78.3  |
> | TextDDI (input length=256)                     | 88.2 (+2.9)    | 88.4 (+2.8)    | 80.3 (+2.0) |
> | w/o information selector (input length=512)    | 85.6     | 85.4     | 78.1  |
> | TextDDI (input length=512)                     | 88.7  (+3.1)   | 88.2 (+2.8)   | 80.7 (+2.6) |
>
> **The additional experimental results *w/o information selector (input length=512)* demonstrate that the information selector continues to provide nearly the same improvement(2.0 ~ 3.6%) on both datasets when the input length is 512. This further validates the contribution of the information selector.**
>
>
>
>
>
> **Q4**: *Most language models the authors utilize are far smaller than the current so-called LLMs. I would suggest not using the term large language models.*
>
> **A4**:
> Thank you very much for this advice, and we will change "large language models (LLMs)" to "pretrained language models (PLMs)".

---

### Meta-Review · Area_Chair_yxQj · 2023-09-17

**Recommendation:** 4

**Metareview:**

This paper investigates a LLM-based approach for predicting drug-drug  interactions (DDI). The key idea of the proposed model is to mutually learn adequate prompts that leverage descriptive information from external resources, in the one hand, and DDI prediction score using a reinforcement learning policy in the other hand.

The reviewers agree that the overall contribution is interesting and innovative, presenting a new research angle on a well-known prediction task for health, namely DDI.
Even if the evaluation part looks reasonable, there have been some concerns about the limited input length, as well as the lack of appropriate ablation study scenarios to fully support the soundness of the results (e.g., DDI predictor / and w/ textual features such as name, truncated vs. full description).

Overall, the discussion phase among the reviewers expressed a positive feeling about the paper and its potential and even its generalizability to other tasks.

---

### Decision · Program_Chairs · 2023-10-07

**Decision:**

Accept-Main

**Comment:**

This paper investigates a LLM-based approach for predicting drug-drug  interactions (DDI). The key idea of the proposed model is to mutually learn adequate prompts that leverage descriptive information from external resources, in the one hand, and DDI prediction score using a reinforcement learning policy in the other hand.

The reviewers agree that the overall contribution is interesting and innovative, presenting a new research angle on a well-known prediction task for health, namely DDI.
Even if the evaluation part looks reasonable, there have been some concerns about the limited input length, as well as the lack of appropriate ablation study scenarios to fully support the soundness of the results (e.g., DDI predictor / and w/ textual features such as name, truncated vs. full description).

Overall, the discussion phase among the reviewers expressed a positive feeling about the paper and its potential and even its generalizability to other tasks.